# Special care dentistry perception among dentists in Jakarta: An online survey study

**Masita Mandasari** [1,2]*, **Febrina Rahmayanti**[2], **Hajer Derbi**[3], **Yuniardini S. Wimardhani**[2]

**1** Oral Medicine Residency Program, Faculty of Dentistry Universitas Indonesia, Jakarta, Indonesia, **2** Oral Medicine Department, Faculty of Dentistry Universitas Indonesia, Jakarta, Indonesia, **3** DCD Special Needs Dentistry Program, Melbourne Dental School, The University of Melbourne, Victoria, Australia

* masitamandasari@ui.ac.id

**Data Availability Statement:** Data set is available in public repository (doi:10.5061/dryad.vhhmgqnsg).

**Funding:** MM received fundings from Universitas Indonesia through International-Indexed

## Abstract

Special Care Dentistry (SCD) or Special Needs Dentistry is a branch of dentistry concerned with the oral health of people with a variety of medical conditions or limitations that require more than routine delivery of care. There were reports on oral status of special care patients and special interest group for SCD dentists in Indonesia has existed. However, there was not perception report on SCD amongst dentists in Jakarta. This paper will describe the perception of dentists in Jakarta towards SCD. A cross-sectional questionnaire, translated and cross adapted to Indonesian, was distributed online through Whatsapp to dentists registered in Jakarta late 2019. Quantitative data was analyzed using statistical software for proportion and correlation using Chi-Square test. The questionnaire explored dentists' perception towards SCD. A total of 250 dentists participated in this study, of them 173 general practitioners and 77 specialist dentists. Most respondents reported that they did not have SCD component during undergraduate dental school and did not provide treatment to patients with special needs in their clinical practice. Most respondents have poor perception of SCD, however, most of the respondents showed motivation and interest towards SCD training. Dentists in Jakarta involved in this study had poor perception of SCD. More efforts should be performed to improve SCD education and awareness.

## Introduction

Oral health was reported to be a significant health problem in Indonesian population [1]. Special Care Patients (SCPs) are in particular more vulnerable to oral health problems than those without, being reported to have higher plaque scores and caries prevalence, as well as poor periodontal health [2, 3]. Maintaining good oral hygiene and access to dental care amongst this cohort was, and still is, often non-existent, especially when compared to the general population. Problems such as lack of oral health awareness amongst caregivers and/or patients, limitation in the ability to perform oral hygiene practices, agility, and poor motor coordination, diet problems, medications' side effects, limited access to dental care, or needing assistance in performing oral hygiene regime are evident to be responsible for the poor oral health in people with disability [4, 5]. The WHO estimated that roughly 15% of world population has some

Publication Grant (PUTI Universitas Indonesia NKB-1921/UN2.RST/HKP.05.00/2020) https://ui. ac.id/en/. The funders had no role in study design, data collection and analysis, decision to publish, or preparation of the manuscript.

**Competing interests:** The authors have declared that no competing interests exist.

kind of disability which translated to over one billion people and this number is projected to be increasing in the future, partially due to the ageing societies and increase in chronic health conditions [6]. According to the Ministry of Health Indonesia 2014 report, it is estimated that 2.45% of Indonesian population are living with disability [7]. With a total population of Indonesia around 270 million in 2019 [8] accompanied by the current advanced medical care, improved standard of living, healthy lifestyle initiatives and increased life expectancy, there are concerns about the capacity to provide adequate dental services for the elderly as their numbers are increasing and that they are retaining more teeth [9].

Special Care Dentistry (SCD) or Special Needs Dentistry (SND) is a branch of dentistry concerned with the oral health of people with a variety of medical conditions or limitations that require more than routine delivery of care [10]. The terms 'Special Needs Dentistry' and 'Special Care Dentistry' are used interchangeably on a global level [9]. Not only that there is discrepancy in its nomenclature, but also variation in definition among different countries and even among professional dental societies. Australia and New Zealand use "Special Needs Dentistry" while Europe, the UK, and the US use "Special Care Dentistry" [11, 12]. In a broader term, SCD focuses on the clinicians' caring for the patients, whereas SND focuses on the patients' needs rather than the care given. The Special Care Dentistry Association (SCDA) defined SCD as "branch of dentistry which provides oral care services for people with physical, medical, developmental, or cognitive conditions, which limit their ability to receive routine dental care" [13]. The term SCD will be used onward in this paper.

Special Care Dentistry has been officially recognized as a specialty in Brazil, United Kingdom, Malaysia, New Zealand, and Australia [11, 14–17]. In Australia and Malaysia, the number of dentists specializing in SCD are scarce thus it can be assumed that SCPs are often treated by general practitioners [9, 18]. It is evident that despite the number of training programs in SCD becoming available around the world, there will still be some time before there are sufficient numbers to cater to patients demand. Therefore, it is imperative to improve dental trainings so graduates can meet some of the demand [19, 20].

Jakarta is the capital city of Indonesia and according to the Indonesian Dentist Association, has the highest number of dentists in Indonesia, including general practitioners and dentists in various specialties. Special care dentistry has not been recognized as a specialty in Indonesia yet, however, the Indonesian Dentist Association has acknowledged the Indonesian Society of Special Care Dentistry (ISSCD) which is a special interest group gathering dentists who have interest in SCD [21] and currently has a relatively small number of members. It can be inferred that in Indonesia, this patient cohort would predominately be managed by the general dentist, making a study into aspects of their perception with this patient group pertinent. This paper will describe the perception of dentists in Jakarta towards SCD.

## Methods

The present study employed a previously validated survey format [9]. This cross-sectional study was conducted online to investigate the perception of dentists in Jakarta towards SCD using cross-adapted questionnaire targeting dentists who were registered members of Indonesian Dental Association in five cities of Jakarta Special Capital Region in their respective Whatsapp chat groups (East Jakarta, South Jakarta, West Jakarta, North Jakarta, and Central Jakarta). The instant messaging application Whatsapp was ubiquitously used in Indonesia. The administrator of each group chat helped to deliver the questionnaire invitation and the survey participation reminders on the third and fifth day after the initial invitation. The data collection of this survey was conducted for seven days in the late 2019. Dentists registered as members of Indonesian Society of Special Care Dentistry, received dental qualification from

foreign dental institutions, and those not practicing dentistry were excluded from this study. Respondent's participation was voluntary, self-recruit, and confidential.

The questionnaire was cross-adapted to Indonesian language according to the method published by Beaton, et al. [22] from the original questionnaire [9] which was in English. The questionnaire contained questions with yes-or-no and open-ended answers, and Likert Scale items. There were six aspects consisting of demographic characteristics, perception of SCD, perceptions of special needs patients, government programs and initiatives for dentistry in SCD, training programs for SCD, and referral of SCPs to specialist dentists. The questionnaire was modified to include additional items regarding awareness of ISSCD. Based on our consultation with ISSCD, neither "Special Care Dentistry" nor "Special Needs Dentistry" were translated into Indonesian since there has not been a consensus for translation/equivalent terminology. The questionnaire used in this study can be accessed from the public repository.

Before the conduction of the main study, the questionnaire was tested for validity and reliability analysis in small group of respondents (N = 33). Validity was tested using face to face validity and reported good comprehension among respondents. Internal reliability was assessed using Cronbach's alpha on questionnaire items using Likert scale. The Cronbach's alpha scores for each item was >0.7 so it can be interpreted that the item was reliable. Retest reliability was assessed using intraclass correlation coefficient obtaining score of 0.818 indicating good reliability.

Statistical analysis of quantitative data was performed using IBM SPSS software (IBM Corp, Armonk, New York, USA). Univariate analysis described the proportion of each variables and bivariate analysis described correlations between variables using Chi-Square test. This study was reviewed and approved by the Ethical Committee Faculty of Dentistry Universitas Indonesia (No. 106/Ethical Approval/FKGUI/IX/2019) and acknowledged by the Indonesian Dental Society in DKI Jakarta and ISSCD. The online questionnaire included informed consent page and only respondents who agreed to participate would continue to complete the questionnaire. The responses were recorded electronically as text.

## Results

### Demographic characteristics

The survey was delivered to 1231 dentists in Jakarta. A total of 250 respondents completed the survey which made the response rate 20.3%. As many as 173 (69.2%) respondents were general practitioner dentists and 77 (30.8%) were specialist dentists. The majority of respondents were aged between 24–71 years with the median age of 33. Most respondents were female (83.2%) and had been practicing dentistry for 5–10 years (40.8%). Among specialist dentists, there were 16 (20.8%) orthodontists, 5 (6.5%) oral medicine specialists, 15 (19.5%) pediatric dentistry specialists, 19 (24.7%) endodontist/conservative dentistry specialists, 3 (3.9%) periodontists, 16 (20.8) prosthodontists, and 3 (3.9%) oral and maxillofacial surgeon.

### Perception of special care dentistry

The majority of respondents (N = 163, 65.2%) reported that they did not receive SCD component during their undergraduate study. Among those respondents who reported that they had SCD component (N = 87, 34.8%), the majority (N = 70, 80.5%) rated themselves as competent in SCD, ranging from "somewhat competent" to "completely competent". There was no significant association between the university at which undergraduate training was obtained nor the year of graduation with regards to the SND training.

Over half of the respondents (N = 140, 56%) claimed that they were able to define SCD. Of those respondents who said yes, there were asked to provide the definition of SCD. If the

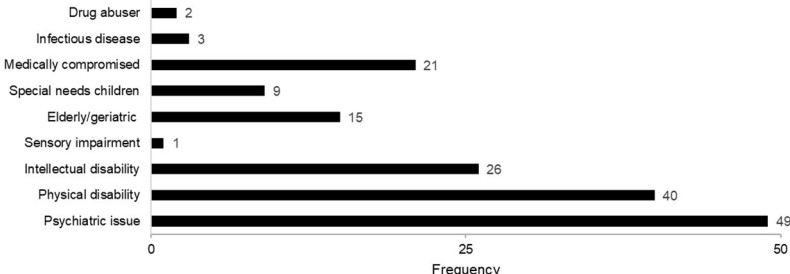

**Fig 1. Frequency of SCD patients.** Respondent who wrote at least one group of SCD patient was considered to have good perception of SCD. Each mention of SCD patient group was counted as one.

respondent mentioned any group of patients who might need SCD (e.g., patients with physical disability, patients with intellectual disability, patients with medical condition, psychiatric issues) the respondent was classified as having 'adequate' knowledge in the SCD field and good SCD perception. Respondents who only translated SCD or SND into Indonesian language or did not provide any answer were classified into groups of poor SCD perception. Fig 1 shows the frequency of SCD patient groups mentioned by respondents. Only 104 (41.6%) respondents were able to define SCD, patients with psychiatric issues was mentioned the most (N = 49) by respondents while patients with sensory impairments was only mentioned once. Medically compromised patients were mentioned by 21 respondents.

A correlation statistical analysis using Chi-Square test showed that specialist dentists demonstrated good SCD perception than general practitioners (Table 1). Other respondents' profiles such as gender, how long they have been practicing as dentists, and the year of their graduation did not show any correlation with SCD perception.

The majority of participants (72%) admitted of not performing SCD in their clinical practices. The most common reason provided by respondents (68.9%) was lack of experience in treating SCPs. Some respondents (31.1%) reporting inadequate surgical facilities (number of surgery or type of facilities) as the reason of difficulty to provide access or treatment to this cohort. Others barriers to providing care included inadequate staff to cater for the special needs of such patients (23.9%), difficulty in managing behavioral problems associated with special care patients (21.7%), more time consuming (13.3%), no interest in treating special care patients or accessibility of the dental surgery (e.g. first floor practice) (9.4%), more staff time is needed (8.3%), high incidence of cancellation or broken appointments (4.4%), and others stated that they had never have SCPs coming to their dental surgery (Table 2).

Respondents were given an opportunity to write their opinions in regards to SCD. Several translated responses in the free-text included the following:

**Table 1. Distribution of dentist competency and SCD perception.**

| SND Perception | Dentistry Competency (N, %) | | Total, N (%) | *p* |
|---|---|---|---|---|
| | General Practioner | Specialist | | |
| **Poor** | 110 (63.6%) | 36 (46.8%) | 146 (58.4) | 0.019* |
| **Good** | 63 (36.4%) | 41 (53.2%) | 104 (41.6) | |

Chi-Square test

*\*p <0.05*

**Table 2. Distribution of reason for not practicing SCD.**

| Answer | Frequency (N = 180) | % |
|---|---|---|
| Lack of experience in treating special care patients | 124 | 68.9 |
| No interest in treating special care patients | 17 | 9.4 |
| Inadequate staff to cater for the special needs of such patients | 43 | 23.9 |
| More staff time is needed | 15 | 8.3 |
| Treating special care patients is more time consuming | 24 | 13.3 |
| Too difficult to manage behavioral problems associated with special care patients | 39 | 21.7 |
| High incidence of cancellation or broken appointments | 8 | 4.4 |
| Inadequate surgical facilities (number of surgery or type of facilities) | 56 | 31.1 |
| Accessibility of the dental surgery (e.g. first floor practice) | 17 | 9.4 |
| Other | 19 | 10.6 |

- "There should be more introduction to SCD and its organization."

- "I didn't know about SCD before I participated in this survey."

- "There should be more training or continuing education programs on SCD

- "Special care dentistry is important so dentist can provide care for patients who need SCD."

- "Special care dentistry should be considered as a specialty or multidisciplinary collaboration."

- "I thought that SCD was part of pediatric dentistry."

## Dentist' perception towards providing treatment to different categories of SCD patients

Participants were asked to rate their perceived level of feeling and comfort in treating different groups of SCD patients. A majority of the participants felt positively in providing treatment to almost every groups of SCD patients except patients with infectious disease followed by patients with behavioral problem (Table 3). Those who treated elderly patients reported feeling "positive" (N = 153, 61.2%). In terms of those who treated physically disabled patients, most rated their experiences either "positive" (N = 159, 63.6%) or "very positive" (N = 44, 17.6%). When asked about their feelings in treating patient with intellectual disability and medically compromised patients, the majority of participants rated their experiences to be "positive" (N = 133, 53.2% and N = 121, 48.4%, respectively) or "very positive" (N = 36, 14.4% and N = 31, 12.4%, respectively). In contrast, treating patients with infectious disease or treating patients with psychiatric or behavioral issues resulted in participants rating as either "negative" (N = 99, 39.6% and N = 86, 34,4%, respectively) or "positive" (N = 73, 29.2% and N = 92, 36,8%, respectively).

**Table 3. Distribution of respondents' perception of feel towards special care patients (N, %).**

| Special Care Patients | Extremely negative | Very negative | Negative | Positive | Very positive | Extremely positive |
|---|---|---|---|---|---|---|
| **Elderly** | 1 (0.4) | 0 (0) | 29 (22.6) | 153 (61.2) | 46 (18.4) | 21 (8.4) |
| **Physical disability** | 1 (0.4) | 2 (0.8) | 28 (11.2) | 159 (63.6) | 44 (17.6) | 16 (6.4) |
| **Intellectual disability** | 3 (1.2) | 12 (4.8) | 54 (21.6) | 133 (53.2) | 36 (14.4) | 12 (4.8) |
| **Medically-complex problem** | 6 (2.4) | 11 (4.4) | 66 (26.4) | 121 (48.4) | 31 (12.4) | 15 (6) |
| **Infectious disease** | 21 (8.4) | 31 (12.4) | 99 (39.6) | 73 (29.2) | 17 (6.8) | 9 (3.6) |
| **Behavior or psychological problem** | 11 (4,4) | 24 (9,6) | 86 (34,4) | 92 (36,8) | 28 (11,2) | 9 (3,6) |

**Table 4. Distribution of respondents' perception of comfort towards special care patients (N, %).**

| Special Care Patients | Extremely uncomfortable | Very uncomfortable | Uncomfortable | Comfortable | Very comfortable | Extremely comfortable |
|---|---|---|---|---|---|---|
| Elderly | 1 (0.4) | 1 (0.4) | 34 (13.6) | 176 (70.4) | 29 (11.6) | 9 (3.6) |
| Physical disability | 1 (0.4) | 3 (1.2) | 43 (17.2) | 169 (67.6) | 25 (10) | 9 (3.6) |
| Intellectual disability | 3 (1.2) | 11 (4.4) | 86 (34.4) | 135 (54) | 11 (4.4) | 4 (1.6) |
| Medically-complex problem | 9 (3.6) | 14 (5.6) | 91 (36.4) | 121 (48.4) | 14 (5.6) | 1 (0.4) |
| Infectious disease | 38 (15.2) | 42 (16.8) | 119 (47.6) | 44 (17.6) | 7 (2.8) | 0 (0) |
| Behavior or psychological problem | 18 (7.2) | 30 (12) | 114 (45.6) | 76 (30.4) | 11 (4.4) | 1 (0.4) |

In terms of comfort (Table 4), most participants felt "comfortable" in treating elderly patients (N = 176, 70.4%), patients with physical disability (N = 169, 67.6%), intellectual disability (N = 135, 54%), and patients with complex medical problems (N = 121, 48.4%). However, most participants reported "uncomfortable" towards patients with infectious disease (N = 119, 47.6%) and patients with behavior or psychological problem (N = 114, 45.6%).

### Government programs and initiatives for special care dentistry

Only a small number (N = 5, 2%) of respondents answered that they knew about government program and initiatives for SCD and mentioned programs such as priority health services for elderly and people with disability in public health center or government's hospital and financial support through the national health insurance. Of those who knew of such programs, they reported that they thought the programs were advantageous for SCPs.

### Training programs for special care dentistry

Almost all participant dentists (N = 243, 97.2%) felt that undergraduate dental students should receive didactic and clinical training in SCD and it should be given in their third year (42%) and fourth year (35.8%). Most respondents showed their willingness in attending continuing education programs in the topic of SCD (89.6%) and would consider attending postgraduate training in SCD (79.6%).

### Referring special care patients to special care dentistry specialist

Table 5 showed the distribution of respondents' criteria in referring patient to SCD specialist. Their responses were almost similarly distributed for every criterion. However, 37 (14.8%) respondents chose that they would always refer SCP to a specialist when they need second opinion, followed by when they were treating patients with complex medical problems or medically compromised patients (N = 35, 14%).

### Discussion

Many studies reported that oral disease and unmet dental needs were more prevalent among SCPs [2, 3]. A significant number of barriers are faced by those patients with special needs. Amongst the most reported barriers in obtaining appropriate level of oral health care are economic burden and physical barriers [23]. Their disability and complicated health care needs may often prevent them from getting adequate dental care [24–26]. The lack of availability of dental practitioners is also a perceived barrier in accessing oral health care service [27, 28].

The use of Whatsapp as a method of recruitment for online epidemiological survey has been reported before [29]. Recruiting respondents by email is more common than Whatsapp in online survey, however both method were reported to have similar recruitment rate which

**Table 5. Distribution of special care patient referral criteria.**

| Criteria | Frequency (N,%) | | | | | |
|---|---|---|---|---|---|---|
| | **Always** | **Very often** | **Often** | **Sometimes** | **Almost never** | **Never** |
| **I would like a second opinion** | 37 (14.8) | 26 (10.4) | 71 (28.4) | 87 (34.8) | 20 (8) | 9 (3.6) |
| **I am uncomfortable performing the necessary procedure** | 22 (8.8) | 21 (8.4) | 78 (31.2) | 89 (35.6) | 27 (10.8) | 13 (5.2) |
| **I am unsure how to proceed with treatment for patient who is medically complex or compromised** | 35 (14) | 34 (13.6) | 76 (30.4) | 81 (32.4) | 17 (6.8) | 7 (2.8) |
| **Patient is intellectually disabled and unable to co-operate** | 28 (11.2) | 28 (11.2) | 74 (29.6) | 76 (30.4) | 29 (11.6) | 15 (6) |
| **Patient is physically disabled and I have no facilities** | 29 (11.6) | 24 (9.6) | 75 (30) | 78 (31.2) | 28 (11.2) | 16 (6.4) |
| **Patient has psychological problem which precludes treatment in a general dental surgery** | 27 (10.8) | 26 (10.4) | 71 (28.4) | 80 (32) | 27 (10.8) | 19 (7.6) |
| **Patient has behavioral problem which makes treatment delivery difficult** | 30 (12) | 31 (12.4) | 76 (30.4) | 73 (29.2) | 26 (10.4) | 14 (5.6) |

were 28.1% and 24.9% respectively [29]. The response rate in this study was 20.3%, being quite similar with the previous published study which was conducted by email [29].

Since SCD has not been considered as a dental specialty in Indonesia, we hypothesized that specialist dentists were more "familiar" with SCD since dental specialist training would include dental management of complex cases like SCD patient. This study found that specialist dentists tend to have good perception towards SCD. However, the majority of the respondents were general practitioners. In addition, more than half of the respondents in this study reported that they did not received SCD component during undergraduate program. Moreover, medically compromised patient was only recognized as part of SCD by 21 respondents. The undergraduate dentistry curriculum, however, according to the Indonesian Dentist Standard must include component of dental management of patients with complex medical health and patients with mental health issues [30]. This curriculum has been applied nationwide since 2011. Thus, this study also showed that most of the respondents were not aware that management of dental patients with complex medical condition is within the scope of SCD.

General dental practitioners have been trained in some tertiary dental curricula to provide care for special needs patients whose spectrum of need is less complicated [11, 31]. However, they often find treating patients with special needs challenging [32] and often report difficulties with communication, thus creating sometimes seemingly insurmountable barriers to access of mainstream dentistry by this population.

It has been estimated that 90% of people requiring SCD can receive treatment in local, primary care centers [33]; however, this was dependent upon adequate education and training of the dental team [20]. Curricula design is the key in achieving the required level of knowledge and skills during undergraduate education [34]. Subjects like Special Needs/Care Dentistry would be the key elements in undergraduate training and ought to be given more emphasize, especially with the increased life expectancy amongst this population [35].

The findings that most respondents in this study cannot described SCD was not completely unexpected since the term SCD itself was relatively new in Indonesia. This result was in parallel with the finding among Malaysian dental students who were unable to define SCD [19]. However, SCD has starting to gain recognition as a specialty in Malaysia [17, 19]. Undergraduate dental curriculum in Malaysia also had included SCD as an integrated module and students were exposed to SCPs through rotation in SCD clinics and visitations to institutions caring for

SCPs [20]. These efforts could also be applied in Indonesia so Indonesian dental students can gain understanding of SCD and later become competent oral care provider to SCPs.

To the best of our knowledge, whether SCD should be a specialty or a special interest group consisted of dentists with different specialty background was still under discussion in Indonesia. The matter of discussion was whether SCD can show the specific knowledge and set of skills which has not existed in any other specialty in dentistry. An editorial article in 2016 argued that SCDA's definition of SCD can be fulfilled by the specialties that have been recognized by the American Dental Association (ADA), as well as by many non-specialty in dentistry [13]. But, no matter with the status of SCD in Indonesia, it did not changed the fact that oral health care needs of SCPs population was being neglected [3, 36]. This is a larger issue that should be addressed.

Most of the study respondents said that they did not perform SCD in their daily practice. Many cited their lack of experience in treating SCPs as the reason, although they generally felt positive and comfortable toward treating SCPs. These results were similar with the results reported in Australia [9]. This study thus provided another evidence that there is an increased need of SCD development among dental professionals.

A recent study about the attitude of dental students toward treating SCPs in Austria showed that implementation of SCD in dental school positively affect students' perception towards SCPs although it did not positively affect their confidence to treat these patients [37]. This should come as a notice to the dental community, especially educators, that SCD component in dental school alone is not necessarily sufficient to prepare students as dentists who can or willing to practice SCD.

It is well reported that dental caries and periodontal disease have a strong correlation with poor oral hygiene. It is not surprising that individuals with special needs were also consistently shown to have more oral disease [2, 3]. In addition, poor oral health including greater restorative procedures and minimal preventive treatment were common amongst those with intellectually disability in the presence or absence of physical disability [26]. Several studies reported that individuals with intellectual disability were more likely to receive more tooth extractions rather than restorations for dental problems, compared with individuals in the general population [38, 39]. Since SCPs highly dependent on their caregivers' perception toward oral health, this could be the reason of the lack of SCPs in dental practice encountered by the participants in this study. It is documented that caregivers of SCP understood that oral care was important, however, there were still some who thought that it was not essential and dental visits should only be performed when the patients were in pain [40]. Preventive measure for this cohort will always require the cooperation, positive attitude, and continued interest and treatment priorities of the caregivers. In addition, caregivers' knowledge of oral hygiene measures impacted on the degree of appropriate oral care provided to their clients. Unfortunately, many caregivers received minimal training in provision of oral care. Lack of supervision and negative attitudes toward dental health by the caregiver have been cited as the obstacles to good oral health [25].

The WHO has published a World Disability Report stated that around 15% or over a billion people in the world have a kind of disability and with the increasing life expectancy of world populations, this number was expected to increase [6]. Therefore, the demand of SCD will also grow in the future and dentists should expect and be prepared to deliver SCD in their dental practice.

Particularly with young SCPs, pediatric dentistry specialists can provide oral health care to this population. However, these patients will reach the age of adulthood and no longer be in the care of pediatric dentists so they could encounter unmet oral care needs [41]. In areas where there were a lack of SCD dentists, SCPs may receive oral and dental care from the

general practitioners [9, 41]. This was no different in the present study especially considering that this specialty is not recognized yet in Indonesia.

The general dentists would rely on their undergrad training or continuous professional development courses attended post-graduation to manage this patient cohort. However, there has been report that not all dentists were willing or able to treat SCPs and this has become the greatest barrier in getting oral healthcare in SCPs [41]. It has also been reported that many respondents in this study would like a second opinion when they treat SCPs. However, there is still no or very minimum treatment pathway for them to follow so far.

Despite the poor perception, most respondents in this study showed interest and motivation in SCD. They expressed their willingness to attend continuing education in SCD. Previous studies showed that dentists will feel more confident and ready to treat SCPs when they had adequate training in their undergraduate training and through continuing education [42, 43]. This is also in accordant to the study of dental students in Austria, that adequate training should also be given to develop confidence [37]. One study also reported that dentists who had positive perception towards their dental education (i.e. satisfied with SCD education), they were more likely to show positive perception in providing oral health care to SCP [42]. With more dentists willing and prepared to treat SCPs, the gap of unmet oral health care can be reduced therefore improving patients' quality of life [44].

Lastly, this study was a self-recruitment study so the result should be cautiously interpreted and not to be generalized. The respondents who recruited themselves into this study were usually those who have interest towards the topic, i.e. SCD, thus they might have given biased responses in the survey. But, this study was an "exploratory" study on a niche topic, therefore a survey that was distributed directly to target population which they could subsequently volunteer to become respondents was a simple, fast, and a cost-efficient method.

## Conclusion

This study explored the perception of SCD amongst the dentists in Jakarta using survey conducted online. Most respondents had poor perception in SCD, reported lack of education in SCD, and did not deliver SCD in their dental practice. This showed that more efforts should be initiated to improve the dental education and to raise the awareness of SCD in Indonesia, not only for dentists but also for the community of SCPs. For starters, it can begin by a collaboration between the Indonesian Dental Council, dental schools, and ISSCD to promote training and education in the area of SCD. Furthermore, this study can be followed up with investigation in the oral health care need of various SCPs to convey the importance of SCD and uncover the barriers to receiving dental care.

## Acknowledgments

Authors would like to thank Dr. Melissa Adiatman for statistical consultation.

## Author Contributions

**Conceptualization:** Masita Mandasari, Febrina Rahmayanti, Yuniardini S. Wimardhani.

**Data curation:** Masita Mandasari.

**Formal analysis:** Masita Mandasari, Febrina Rahmayanti, Hajer Derbi, Yuniardini S. Wimardhani.

**Funding acquisition:** Masita Mandasari, Yuniardini S. Wimardhani.

**Investigation:** Masita Mandasari.

**Methodology:** Masita Mandasari, Febrina Rahmayanti, Yuniardini S. Wimardhani.

**Project administration:** Febrina Rahmayanti, Yuniardini S. Wimardhani.

**Supervision:** Febrina Rahmayanti, Yuniardini S. Wimardhani.

**Visualization:** Masita Mandasari.

**Writing – original draft:** Masita Mandasari, Hajer Derbi.

**Writing – review & editing:** Masita Mandasari, Hajer Derbi.

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
