## [Decision Letter · Decision Letter 0]

28 Jan 2021

PONE-D-20-32506

Special Care Dentistry Perception among Dentists in Jakarta: An Online Survey Study

PLOS ONE

Dear Dr. Mandasari,

Thank you for submitting your manuscript to PLOS ONE. After careful consideration, we feel that it has merit but does not fully meet PLOS ONE’s publication criteria as it currently stands. Therefore, we invite you to submit a revised version of the manuscript that addresses the points raised during the review process.

We look forward to receiving your revised manuscript.

Kind regards,

Frédéric Denis, Ph.D.

Academic Editor

PLOS ONE

Journal Requirements:

Reviewers' comments:

Reviewer's Responses to Questions

**Comments to the Author**

1. Is the manuscript technically sound, and do the data support the conclusions?

Reviewer #1: Yes

Reviewer #2: Yes

2. Has the statistical analysis been performed appropriately and rigorously? 

Reviewer #1: Yes

Reviewer #2: I Don't Know

3. Have the authors made all data underlying the findings in their manuscript fully available?

Reviewer #1: Yes

Reviewer #2: Yes

4. Is the manuscript presented in an intelligible fashion and written in standard English?

Reviewer #1: Yes

Reviewer #2: Yes

5. Review Comments to the Author

Reviewer #1: The manuscript entitled "Special Care Dentistry Perception among Dentists in Jakarta: An Online Survey Study" poses an interesting question highlighting a somewhat neglected, yet important, dentistry field. The limitation is the sample size.

Reviewer #2: Please, find my comments in the attached file (PDF file of the manuscript). All my comments could be seen in the attached file (PDF file of the manuscript). The authors should fulfill all my comments in the manuscript.

6. PLOS authors have the option to publish the peer review history of their article (what does this mean?). If published, this will include your full peer review and any attached files.

Reviewer #1: No

Reviewer #2: No

---

## [Author Response · Author response to Decision Letter 0]

21 Mar 2021

The authors would like to thank the editor and the reviewers for improving the manuscript. We believed that we have addressed the issues raised by the editor and the reviewers and they have been mentioned in the rebuttal letter. The data analyzed for this manuscript has been stored in public repository, available for review by the reviewers and the editor. Thank you.

---

## [Editor Report · Decision Letter 1]

24 Mar 2021

Special Care Dentistry Perception among Dentists in Jakarta: An Online Survey Study

PONE-D-20-32506R1

Dear Dr. Mandasari,

We’re pleased to inform you that your manuscript has been judged scientifically suitable for publication and will be formally accepted for publication once it meets all outstanding technical requirements.

Kind regards,

Frédéric Denis, Ph.D.

Academic Editor

PLOS ONE
---

## [Editor Report · Acceptance letter]

29 Mar 2021

PONE-D-20-32506R1 

Special care dentistry perception among dentists in Jakarta: An online survey study 

Dear Dr. Mandasari:

I'm pleased to inform you that your manuscript has been deemed suitable for publication in PLOS ONE. Congratulations! Your manuscript is now with our production department. 

Kind regards, 

on behalf of

Dr. Frédéric Denis 

Academic Editor

PLOS ONE